# Effect of an antenatal diet and lifestyle intervention and maternal BMI on cord blood DNA methylation in infants of overweight and obese women: The LIMIT Randomised Controlled Trial

Jennie Louise[1]*, Andrea R. Deussen[1], Berthold Koletzko[2], Julie Owens[3], Richard Saffery[4,5], Jodie M. Dodd[1,6]

1 Discipline of Obstetrics & Gynaecology and The Robinson Research Institute, The University of Adelaide, Adelaide, South Australia, Australia, 2 Division of Metabolic and Nutritional Medicine, Dept. Paediatrics, Dr. von Hauner Children's Hospital, LMU—Ludwig-Maximilian-Universität, Munich, Germany, 3 Deputy Vice-Chancellor's Research Office, Deakin University, Geelong, Australia, 4 Epigenetics Group, Murdoch Children's Research Institute, Royal Children's Hospital, Parkville, Victoria, Australia, 5 Department of Paediatrics, University of Melbourne, Parkville, Victoria, Australia, 6 Department of Perinatal Medicine, Women's and Babies Division, The Women's and Children's Hospital, Adelaide, South Australia, Australia

* jennie.louise@adelaide.edu.au

**Data Availability Statement:** The minimal dataset for this study includes participants' personal and

## Abstract

### Background

To investigate the effect of an antenatal diet and lifestyle intervention, and maternal pre-pregnancy overweight or obesity, on infant cord blood DNA methylation.

### Methods

We measured DNA methylation in 645 cord blood samples from participants in the LIMIT study (an antenatal diet and lifestyle intervention for women with early pregnancy BMI $\geq$25.0 kg/m$^2$) using the Illumina 450K BeadChip array, and tested for any differential methylation related to the intervention, and to maternal early pregnancy BMI. We also analysed differential methylation in relation to selected candidate genes.

### Results

No CpG sites were significantly differentially methylated in relation to either the diet and lifestyle intervention, or with maternal early pregnancy BMI. There was no significant differential methylation in any of the selected genes related to the intervention, or to maternal BMI.

### Conclusion

We found no evidence of an effect of either antenatal diet and lifestyle, or of maternal early pregnancy BMI, on cord blood DNA methylation.

health information (age, BMI, parity, smoking status, quintile of socioeconomic disadvantage) along with epigenetic data. Participants consented to specific use of this data for the purposes of research related to the effect of overweight and obesity, and of an antenatal diet and lifestyle intervention, on gene expression in infants. Consent was not sought for unspecified future research (including deposition in a public repository). Under the NHMRC National Statement on Ethical Conduct in Human Research (specifically chapters 2.3 and 3.1), any use of the data not covered by the scope of the original consent requires either that consent be sought from participants, or that a waiver of consent be granted by the ethics committee. As such, it is the determination of the WCHN Human Research Ethics Committee that these data may not be made available in a public repository. To facilitate sharing of data, an established process has been in place since the commencement of the LIMIT study. Data access requests, describing the proposed use(s) of the data, may be made by contacting the Women's and Children's Health Network Human Research Ethics Committee, 72 King William St., North Adelaide, South Australia, 5006 (HealthWCHNResearch@sa.gov.au), and the LIMIT Data Access Committee (University of Adelaide, WCH Campus, 72 King William St, North Adelaide, South Australia 5006 (email ATTN: LIMIT Data Access Committee to limit@adelaide.edu.au). The WCHN HREC are independent of the LIMIT study and are responsible for granting waiver of consent for the proposed research. The LIMIT data access committee will provide access to the data once this approval has been granted.

**Funding:** The LIMIT Randomised Trial was funded by an NHMRC grant (ID519240), awarded to JMD. Funding for the DNA methylation analysis was from the Commission of the European Communities, the 7th Framework Programme, contract FP7-289346-EARLY NUTRITION, awarded to BK. JMD was also supported by NHMRC Practitioner Fellowships (ID627005 and ID1078980) and Investigator Grant (ID1196133). BK is supported by the European Joint Programming Initiative Project NutriPROGRAM and the German Ministry of Education and Research, Berlin (Grant Nr. 01 GI 0825). BK is the Else Kröner Seniorprofessor of Paediatrics at LMU – University of Munich, financially supported by the Else Kröner-Fresenius-Foundation, the LMU Medical Faculty and the LMU University Hospitals. The funders had no role in study design, data collection and analysis, decision to publish, or preparation of the manuscript

## Clinical trials registration

ACTRN12607000161426

## Introduction

There is a well recognised link between maternal overweight and obesity and the risk of overweight and obesity in children. Infants born to women who are overweight or obese in pregnancy have, on average, higher birthweight for gestational age [1, 2], and higher adiposity [2, 3]. They are also recognised to be at greater risk of childhood overweight and obesity [4], and its associated health consequences later in life [5–7]. These transgenerational effects most likely have multiple causes, including environmental exposures and genetic factors, although recent attention has been focused on peri-conceptional or *in-utero* exposures [8]. Such exposures include maternal overweight and obesity, gestational weight gain, antenatal nutrition and physical activity, pregnancy complications (including gestational diabetes (GDM) and hypertension), as well as social and behavioural factors. The mechanisms by which these exposures contribute to an increased susceptibility to child obesity are not fully understood. Current evidence suggests shared genetics explains only a small amount of the heritability of obesity [1, 9, 10]. Other postulated mechanisms include alterations to the maternal gut microbiome [1, 11], maternal hyperinsulinaemia and hyperglycaemia in pregnancy [4, 12]. While there is some evidence of paternal influences, these have been relatively under-studied [7, 12].

Among these possibilities, epigenetic mechanisms have been the focus of much recent investigation. Epigenetics is broadly taken to refer to changes in gene function which occur in the absence of changes to the underlying DNA [7, 11, 12], resulting in changes to regulation and expression of genes via mechanisms such as DNA methylation (DNAm), histone modification, and noncoding RNAs [4, 7]. DNA methylation is the most widely studied epigenetic mechanism, and involves the attachment of a methyl group to a CpG dinucleotide, which is then passed on in DNA replication and cell division [2, 4, 13, 14]. Methylation, particularly in gene promoter regions, is generally believed to contribute to gene silencing [7, 11] although this depends on a number of factors, including the methylation site (e.g. promoter region vs. gene body) [2, 14], and the interaction between different epigenetic mechanisms (e.g. between DNAm and histone modification) [2, 4].

Epigenetic mechanisms have been proposed as a potential means by which maternal obesity predisposes to obesity in offspring, both via the metabolic effects of obesity and via maternal diet in pregnancy (though these two effects are sometimes conflated). Evidence from non-human models demonstrates that maternal diet in pregnancy can alter DNAm profiles of offspring, and that this in turn influences offspring adiposity [2, 4, 7, 10, 11, 15]. The evidence from human studies to date is less robust. Studies of cohorts of women exposed to extreme *under*nutrition, either periconceptionally or in pregnancy [12], have demonstrated effects on DNAm (e.g. on the *IGF2* imprinted gene) in offspring, along with a predisposition to adiposity, type 2 diabetes and other metabolic disorers in later life [2, 7, 12, 14, 16]. A range of studies have also reported differential methylation at various genomic loci in neonatal cord blood associated with pre-pregnancy overweight and obesity (S1 Table), and in children born to women following bariatric surgery [5].

The aims of this prespecified secondary study were to investigate DNAm in cord blood samples from 645 participants in the LIMIT randomised controlled trial of an antenatal diet and lifestyle intervention in women with body mass index (BMI) $\geq$25.0 kg/m$^2$. We undertook

**Competing interests:** The authors have declared that no competing interests exist.

**Abbreviations:** CpG, Cytosine-phosphate-Guanine; DNAm, DNA methylation; BMI, Body mass index; GDM, Gestational Diabetes Mellitus; IGF2, Insulin-like growth factor 2; IGF, Insulin-like growth factor; RXRA, Retinoid x receptor alpha; PPARGC1A, Peroxisome proliferator-activated receptor gamma coactivator 1; MEST, Mesoderm-specific transcript; IQR, Interquartile range; SD, Standard deviation; FC, Fold change; BMIQ, Beta-Mixture Quantile; SWAN, Subset-Within-Array Normalisation.

an epigenome-wide analysis of differential methylation related to the diet and lifestyle intervention, and/or related to maternal early pregnancy BMI. We also investigated differential methylation in selected genes where previous research had found differential methylation associated with maternal BMI, and which were plausibly related to obesity, adiposity, metabolism, or growth, namely:

- *IGF2* on chromosome 11: a maternally expressed imprinted gene, expression of which has been found to relate to circulating IGF in cord blood. DNA methylation in the imprinting region for *IGF2* and the associated paternally-expressed imprinted gene *H19*, have been found to be associated with adiposity [17];

- *RXRA* on chromosome 9: differential methylation of this gene in cord blood has been found to be associated with childhood adiposity [10, 14];

- *PPARGC1A* on chromosome 4: a gene which regulates genes involved in energy metabolism and has been found to be differentially methylated in adults with impaired glucose tolerance and in adults exposed to high-fat overfeeding [8]; some studies have found evidence of differential DNAm in cord blood associated with maternal obesity [18, 19];

- *MEST*, a mostly paternally-expressed imprinted gene which may play a role in adipocyte differentiation, and which has been found to be differentially methylated in cord blood of women with obesity compared to normal BMI, and also of women with GDM [20].

## Methods

### The LIMIT Randomised Controlled Trial

The LIMIT randomised, controlled trial evaluated the effects of an antenatal diet and lifestyle intervention for women with early pregnancy BMI $\geq$25.0 kg/m$^2$, with findings extensively reported elsewhere [21]. Women were eligible if they had early pregnancy BMI $\geq$25.0 kg/m$^2$, a singleton pregnancy between $10^{+0}$ and $20^{+0}$ weeks' gestation, and no previously existing diabetes. A total of 2212 women were randomised to receive either Lifestyle Advice (n = 1108), a comprehensive diet and lifestyle intervention, or Standard Care (n = 1104), in which antenatal care was delivered according to local guidelines (and did not include information on diet or physical activity). The primary outcome was birth of an infant large for gestational age (LGA). While there were no significant differences observed between the groups in relation to this outcome, a significantly lower incidence of birthweight >4kg was observed in the Lifestyle Advice group (Relative Risk (RR) 0.82 (95% Confidence Interval (CI): 0.68, 0.99, p = 0.04). Additionally, measures of diet quality and physical activity were improved in women in the Lifestyle Advice group compared with those in the Standard Care group [22].

Cord Blood DNA for a range of secondary studies was collected at the time of birth from consenting participants, and was frozen as whole blood preserved with EDTA. Funding was available to perform DNA methylation analysis for a total of 649 samples, which were randomly selected from the total number of available samples, balanced between the Lifestyle Advice and Standard Care groups. After DNA extraction, genome-wide DNA methylation was performed using the Illumina Infinium HumanMethylation 450K Bead-Chip array. Results were supplied as raw probe intensities (*idats* files).

**Ethics approval and consent to participate.**   The study was reviewed by the ethics committee of each participating institution including the Women's and Children's Health Network Human Research Ethics Committee (1839 & 2051); The Central and Northern Adelaide Health Network Human Research Ethics Committee (2008033) and the Southern Adelaide

Local Health Network Human Research Ethics Committee (formerly Flinders Clinical Research Ethics Committee) (128/08).

Informed, written consent was obtained for all participants to participate in the LIMIT study, and additional written consent was obtained to collect samples of umbilical cord blood at delivery for the purposes of gene expression research related to weight and to the diet and lifestyle intervention.

## Data processing

Data processing and analysis was performed using R version 4.0 [23]. The *minfi* package [24] was used to read in the raw idats files, and to calculate detection p values (comparison of methylated (M) and unmethylated (U) intensities to background signal) both for all samples (across all probes) and all probes (across all samples). There were a total of 662 sets of results in the data, as 13 samples had been rerun due to chip failure. The initial results for these samples were identified and excluded. These were the only samples classified as 'failed' (detection p value $\geq 0.01$). Of the 649 valid samples, four were excluded because of labelling errors where the correct study identifier could not be ascertained. A total of 645 samples were therefore retained for processing and analysis. The raw data for these 645 samples were converted to $\beta$ values $\left(\frac{M}{M+U+offset}\right)$ and normalisation (removal of technical variation due to, e.g. probe type differences or background signal) was undertaken using the Subset-Quantile Normalisation method [25, 26] as implemented in *minfi*. Following normalisation, failed probes (defined as detection p value $\geq 0.001$ in 25% or more of the 645 samples) were filtered out. Finally, probes identified as cross-reactive [27], probes with an identified SNP within 3 nucleotides of the CpG site and minor allele frequency $>1\%$, and probes on the X and Y chromosomes were filtered out using the *DMRCate* package [28]. This left 426,572 probes available for analysis. Batch effects were not removed at the processing stage [29] but were instead adjusted for in analyses. The *estimateCellCounts* function in *minfi* was used, with Cord Blood reference data, to estimate the proportions of B cells, CD4T, CD8T, granulocytes, monocytes, natural killer and nucleated red blood cells, and these estimated proportions were likewise used for adjustment in the analysis.

To investigate the sensitivity of the analysis results to choice of data processing methods, effects were also estimated using models run using a range of alternative analysis datasets. Firstly, raw data were also normalised using the Beta-Mixture Quantile (BMIQ) method [30] and the Subset-Within-Array Normalisation (SWAN) method [31]. Secondly, datasets were created in which batch effects were handled using the ComBat batch-effect-removal tool [32] implemented in the *ChAMP* package [33] instead of adjustment for batch effects in the models. The results of analyses using these datasets are reported in brief below, but are described in detail in a separate publication.

## Statistical analysis

Statistical analyses of epigenome-wide data were conducted on M-values (logit-transformed $\beta$) using linear models, with adjustment of standard errors using Empirical Bayes methods, as implemented in the *limma* package [34]. The primary analysis model included intervention group (Lifestyle Advice vs Standard Care), BMI (as a continuous variable), their interaction, and the additional covariates parity (0 vs 1+), maternal age (continuous), smoking status, infant sex and study centre. Sample batch and estimated cell type proportions were also included as adjustment variables as described above. The number of probes differentially expressed between Lifestyle Advice and Standard Care groups (at different levels of maternal

BMI), or corresponding to differences in maternal BMI (in each of the intervention groups) were determined using the *decideTests* function in *limma*, using Benjamini-Hochberg method (controlling for a false discovery rate of 5%) to adjust for multiple comparisons.

Secondary sensitivity analyses were also carried out, including unadjusted models (adjusted for only batch and cell type proportion) and models including interactions between intervention and sex, or between BMI and sex (as it has been hypothesised that effects of maternal obesity on gene expression may differ by infant sex [11]).

For candidate gene analyses, all probes at or near (±2000bp) each of the genes of interest were extracted, and linear models were fitted for each probe separately, and for the average M-value across all probes. As above, the models included intervention group, BMI and their interaction, as well as covariates (batch, cell type proportions, parity, maternal age, smoking status, infant sex, study centre, and quintile of relative socioeconomic disadvantage). The mean difference in M-values between Lifestyle Advice and Standard Care groups, and corresponding to a 5-unit increase in maternal BMI, was estimated, along with 95% Confidence Intervals.

## Results

Baseline characteristics of participants whose data is included in this analysis are described in Table 1, and are similar to those of the full LIMIT cohort [21]. The median early pregnancy BMI was 31 kg/m$^2$ (Interquartile Range (IQR) 28–37 kg/m$^2$. A majority of women (60%) were in their second or subsequent pregnancy, and had a mean age of 29 years (SD 5 years). Most (85%) were nonsmokers, and almost all (91%) were of Caucasian ethnicity. Half of the women were from the highest two quintiles of relative socioeconomic disadvantage. Infant sex was evenly divided between males (51%) and females (49%).

### Epigenome-wide analyses

Results of tests for differential methylation associated with the intervention are shown in Table 2 and Fig 1 (top panels). Even using the less strict Benjamini-Hochberg method for Type I error control, there were no probes which were significantly differentially methylated between the Lifestyle Advice and Standard Care groups, and there was no evidence for effect modification by maternal BMI. The top 10 differentially methylated probes by p value were spread across the genome (with the exception of two probes on chr13 mapped to *C13orf34/C13orf37*), and effect sizes were small, (absolute log-FC between 0.1 and 0.15, corresponding to approximately 1.01x higher methylation). The top 10 differentially methylated probes by log-Fold Change (i.e. the probes where the magnitude of difference in methylation was greatest) did not overlap with the top 10 by p-value, and these effect sizes were relatively small (absolute log-FC all being between 0.3 and 0.4, corresponding 1.2 to 1.3 times higher methylation).

Results of tests for differential methylation associated with maternal BMI are shown in Table 3 and Fig 1 (lower panels). There were no probes which demonstrated significant differential methylation according to maternal BMI. As with the intervention effects, the top 10 probes were spread across the genome, with quite small effect sizes. The 10 probes with greatest estimated log-FC did not overlap with those 10 probes with smallest p values.

Results of sensitivity analyses generally confirmed the results of the main analyses. No differentially methylated probes corresponding to intervention or BMI effects were detected in any of the alternative models fitted. There was no evidence of any effects in the unadjusted model (Table 1 in S2 Table), or of effect modification between infant sex and either intervention or BMI.

**Table 1. Baseline characteristics of participants.**

| Characteristic | Lifestyle Advice | Standard Care | Overall |
|---|---|---|---|
| Overall Numbers | n = 325 | n = 320 | n = 645 |
| BMI (kg/m$^2$): Median (IQR) | 31.40 (28.10, 36.20) | 31.45 (27.98, 36.90) | 31.40 (28.00, 36.50) |
| BMI Category: (N%) | | | |
| • 25.0–29.9 | 129 (39.69) | 130 (40.62) | 259 (40.16) |
| • 30.0–34.9 | 99 (30.46) | 86 (26.88) | 185 (28.68) |
| • 35.0–39.9 | 58 (17.85) | 55 (17.19) | 113 (17.52) |
| • ≥40.0 | 39 (12.00) | 49 (15.31) | 88 (13.64) |
| Height(cm): Mean (SD) | 165.29 (6.66) | 164.73 (6.48) | 165.01 (6.57) |
| Weight(kg): Mean (SD) | 89.81 (17.48) | 89.75 (18.65) | 89.78 (18.06) |
| Parity: N(%) | | | |
| • 0 | 141 (43.38) | 128 (40.00) | 269 (41.71) |
| • 1+ | 184 (56.62) | 192 (60.00) | 376 (58.29) |
| Age at TE: Mean (SD) | 29.28 (5.56) | 29.63 (5.24) | 29.45 (5.41) |
| Smoking: N(%) | | | |
| • No | 274 (84.31) | 274 (85.62) | 548 (84.96) |
| • Yes | 47 (14.46) | 37 (11.56) | 84 (13.02) |
| • Missing | 4 (1.23) | 9 (2.81) | 13 (2.02) |
| Ethnicity: N(%) | | | |
| • Non-Caucasian | 29 (8.92) | 29 (9.06) | 58 (8.99) |
| • Caucasian | 294 (90.46) | 291 (90.94) | 585 (90.70) |
| • Missing | 2 (0.62) | 0 (0.00) | 2 (0.31) |
| SEIFA IRSD^ Quintile: N(%) | | | |
| • Q1 | 107 (32.92) | 87 (27.19) | 194 (30.08) |
| • Q2 | 61 (18.77) | 83 (25.94) | 144 (22.33) |
| • Q3 | 59 (18.15) | 52 (16.25) | 111 (17.21) |
| • Q4 | 46 (14.15) | 52 (16.25) | 98 (15.19) |
| • Q5 | 52 (16.00) | 46 (14.37) | 98 (15.19) |
| Infant Sex: N(%) | | | |
| • Male | 164 (50.46) | 163 (50.94) | 327 (50.70) |
| • Female | 161 (49.54) | 157 (49.06) | 318 (49.30) |
| Study Site: N(%) | | | |
| • WCH | 135 (41.54) | 136 (42.50) | 271 (42.02) |
| • FMC | 98 (30.15) | 103 (32.19) | 201 (31.16) |
| • LMH | 92 (28.31) | 81 (25.31) | 173 (26.82) |

SD = standard deviation

IQR = interquartile range

IRSD = Socioeconomic index as measured by SEIFA Index of Relative Socio-economic Disadvantage[35]

In data normalised using different methods, the overall results were generally similar to the main analysis. In data normalised using the SWAN method. there were no significantly differentially methylated probes corresponding to intervention effects, BMI effects, or their interaction (Table 2 in S2 Table). In BMIQ-normalised data, no differentially methylated probes were found for intervention effects, or for the effect of BMI in the Lifestyle Advice group; 5 probes were significantly differentially methylated for the effect of BMI in the Standard Care group (Table 3 in S2 Table). Where a supervised ComBat algorithm (specifying Intervention, BMI and their interaction as effects of interest) was used (in the SQN normalised data) instead of

**Table 2. Top 10 differentially methylated probes (lifestyle advice vs standard care).**

| Rank | Top 10 Probes by p-Value | | | | | Top 10 Probes by log-Fold Change | | | | |
|---|---|---|---|---|---|---|---|---|---|---|
| | chr | Name | UCSC RefGene Name | logFC (95% CI)^ | adj P.Val* | chr | Name | UCSC RefGene Name | logFC (95% CI)^ | adj P.Val* |
| | | | | | Intervention at Mean BMI[a] | | | | | |
| 1 | chr19 | cg03057840 | | -0.10 (-0.14, -0.06) | 0.26 | chr17 | cg08103988 | | 0.50 (0.16, 0.84) | >0.99 |
| 2 | chr4 | cg14712262 | *ZFYVE28* | -0.06 (-0.08, -0.03) | >0.99 | chr17 | cg24686902 | | 0.44 (0.13, 0.75) | >0.99 |
| 3 | chr13 | cg20260570 | *C13orf34;C13orf37* | -0.08 (-0.12, -0.04) | >0.99 | chr17 | cg21358336 | | 0.43 (0.14, 0.73) | >0.99 |
| 4 | chr21 | cg01233397 | | 0.11 (0.06, 0.16) | >0.99 | chr1 | cg04798314 | *SMYD3* | -0.30 (-0.67, 0.07) | >0.99 |
| 5 | chr12 | cg09636302 | *HAL* | -0.11 (-0.16, -0.06) | >0.99 | chr17 | cg08750459 | | 0.29 (0.09, 0.49) | >0.99 |
| 6 | chr18 | cg17242353 | | 0.14 (0.08, 0.21) | >0.99 | chr1 | cg06928484 | *VANGL2* | -0.28 (-0.53, -0.04) | >0.99 |
| 7 | chr11 | cg13932624 | *TBRG1* | -0.06 (-0.08, -0.03) | >0.99 | chr2 | cg04131969 | *MYADML* | -0.28 (-0.70, 0.15) | >0.99 |
| 8 | chr4 | cg16269431 | *GLRB* | -0.06 (-0.09, -0.03) | >0.99 | chr1 | cg08477332 | *S100A14* | -0.28 (-0.52, -0.03) | >0.99 |
| 9 | chr12 | cg11551902 | *FOXM1;C12orf32* | -0.14 (-0.21, -0.07) | >0.99 | chr10 | cg02113055 | | 0.27 (-0.11, 0.64) | >0.99 |
| 10 | chr8 | cg24258108 | *WHSC1L1* | 0.10 (0.05, 0.15) | >0.99 | chr19 | cg25755428 | *MRI1* | 0.26 (0.00, 052) | >0.99 |
| | | | | | Intervention at +5 BMI[b] | | | | | |
| 1 | chr19 | cg03057840 | | -0.09 (-0.13, -0.05) | >0.99 | chr17 | cg08103988 | | 0.51 (0.17, 0.85) | >0.99 |
| 2 | chr12 | cg09636302 | *HAL* | -0.12 (-0.17, -0.07) | >0.99 | chr17 | cg24686902 | | 0.45 (0.14, 0.76) | >0.99 |
| 3 | chr21 | cg01233397 | | 0.11 (0.06, 0.16) | >0.99 | chr17 | cg21358336 | | 0.44 (0.14, 0.74) | >0.99 |
| 4 | chr4 | cg14712262 | *ZFYVE28* | -0.06 (-0.08, -0.03) | >0.99 | chr1 | cg08477332 | *S100A14* | -0.31 (-0.57, -0.06) | >0.99 |
| 5 | chr13 | cg20260570 | *C13orf34;C13orf37* | -0.08 (-0.11, -0.04) | >0.99 | chr1 | cg04798314 | *SMYD3* | -0.30 (-0.67, 0.07) | >0.99 |
| 6 | chr18 | cg17242353 | | 0.14 (0.8, 0.21) | >0.99 | chr17 | cg08750459 | | 0.30 (0.09, 0.50) | >0.99 |
| 7 | chr16 | cg06730286 | *IFT140* | -0.08 (-0.12, -0.04) | > 0.99 | chr8 | cg24634471 | *JRK* | -0.29 (-0.57, 0.00) | >0.99 |
| 8 | chr20 | cg11336672 | *RBL1* | 0.14 (0.07, 0.21) | >0.99 | chr1 | cg06928484 | *VANGL2* | -0.28 (-0.53, -0.03) | >0.99 |
| 9 | chr11 | cg13932624 | *TBRG1* | -0.06 (-0.08, -0.03) | >0.99 | chr10 | cg02113055 | | 0.28 (-0.10, 0.66) | >0.99 |
| 10 | chr17 | cg04435975 | *LOC404266;HOXB6* | 0.08 (0.04, 0.11) | >0.99 | chr2 | cg04131969 | *MYADML* | -0.28 (-0.71, 0.16) | >0.99 |

* Adjusted for multiple comparisons using Benjamini-Hochberg method

^ Model included as covariates parity (0 vs 1+), age (continuous), smoking status, quintile of socioeconomic disadvantage, study centre, infant sex, array batch and estimated cell type proportions (BCell, CD4T, CD8T, Granulocytes, Monocytes, NK, nRBC).

[a] Effect of intervention estimated at the mean BMI for the cohort (approx. 30 kg/m$^2$)

[b] Effect of intervention estimated at mean + 5 kg/m$^2$ BMI (approx 35 kg/m$^2$)

correction for batch in the analysis model, several probes were found to be differentially methylated for most of the effects (Table 4 in S2 Table). However, none of the probes which were significantly differentially methylated in one analysis were replicated in another; the 5 significant probes in BMIQ-normalised data did not even appear in the top-ranked probes in SQN or SWAN-normalised data.

## Candidate gene analysis

The results of candidate gene analyses are presented in Figs 1–4 in S1 File. There was no evidence of differential methylation of probes mapped to *PPARGC1A*, *IGF2*, *RXRA*, or *MEST*, related to either intervention or maternal BMI. For all genes, the pattern of methylation across all probes was similar between the Standard Care and Lifestyle Advice group, and between different maternal BMI values. Estimated effects did not have a consistent direction for either intervention or BMI, with a combination of positive and negative effect estimates across probes. While there were a few individual probes where effects were statistically significant (p<0.05), these p values were not adjusted for multiple comparisons, and it is doubtful that they are meaningful in the context of a large number of other probes in which no effects were evident.

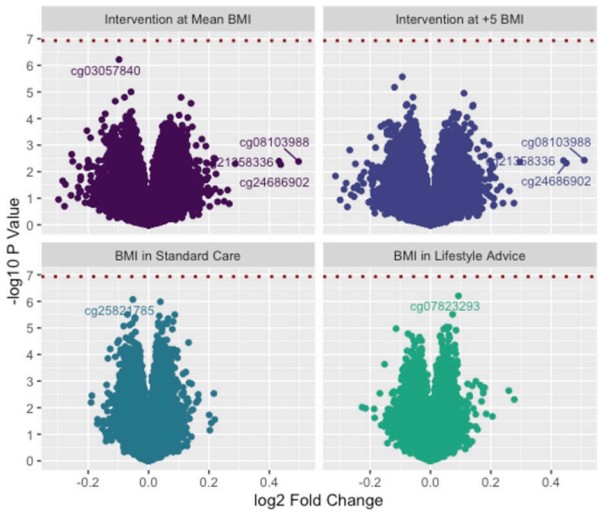

**Fig 1. Volcano plots (-log10 p value vs log2 fold change) for intervention effects (at mean BMI and at +5 kg/m2 BMI) and BMI effects (in lifestyle advice and standard care groups).**

**Table 3. Top 10 differentially methylated probes (5 kg/m2 increase in BMI).**

| Rank | Top 10 Probes by p-Value | | | | | Top 10 Probes by log-Fold Change | | | | |
|---|---|---|---|---|---|---|---|---|---|---|
| | chr | Name | UCSC RefGene Name | logFC (95% CI)^ | adj P.Val | chr | Name | UCSC RefGene Name | logFC (95% CI)^ | adj P.Val |
| | | | | | BMI in Standard Care Group | | | | | |
| 1 | chr3 | cg25821785 | CACNA2D2 | -0.05 (-0.07, -0.03) | 0.22 | chr6 | cg06864789 | | 0.03 (0.00, 0.06) | 0.69 |
| 2 | chr10 | cg21348752 | C10orf114;MIR1915 | 0.04 (0.02, 0.05) | 0.22 | chr17 | cg03226844 | RPH3AL | -0.03 (-0.05, -0.01) | 0.57 |
| 3 | chr3 | cg01919208 | LAMB2 | -0.07 (-0.10, -0.04) | 0.23 | chr6 | cg18136963 | | 0.03 (0.01, 0.06) | 0.71 |
| 4 | chr10 | cg18646207 | VAX1 | 0.09 (0.05, 0.12) | 0.23 | chr8 | cg03547562 | | 0.03 (0.01, 0.05) | 0.76 |
| 5 | chr9 | cg01263574 | TMEM8C | 0.04 (0.02, 0.05) | 0.23 | chr21 | cg11287055 | DSCR3 | -0.03 (-0.05, -0.01) | 0.66 |
| 6 | chr10 | cg16310045 | TCF7L2 | -0.05 (-0.06, -0.03) | 0.23 | chr9 | cg13558371 | CRB2 | -0.03 (-0.05, -0.01) | 0.61 |
| 7 | chr2 | cg16639766 | HJURP | 0.06 (0.04, 0.09) | 0.23 | chr3 | cg03329597 | MYH15 | -0.03 (-0.05, 0.00) | 0.58 |
| 8 | chr7 | cg22005393 | DNAJC2 | -0.05 (-0.06, -0.03) | 0.23 | chr1 | cg01072550 | | -0.03 (-0.04, -0.01) | 0.72 |
| 9 | chr2 | cg05223061 | NGEF | 0.08 (0.05, 0.12) | 0.23 | chr11 | cg24851651 | CCS | 0.03 (0.00, 0.05) | 0.69 |
| 10 | chr6 | cg27244242 | LY6G5C | 0.03 (0.02, 0.05) | 0.23 | chr13 | cg20293942 | | 0.03 (0.00, 0.05) | 0.60 |
| | | | | | BMI in Lifestyle Advice Group | | | | | |
| 1 | chr11 | cg07823293 | TBRG1 | 0.09 (0.06, 0.13) | 0.27 | chr6 | cg06864789 | | 0.28 (0.08, 0.47) | >0.99 |
| 2 | chr4 | cg12630714 | | 0.07 (0.04, 0.10) | 0.63 | chr6 | cg18136963 | | 0.26 (0.09, 0.43) | >0.99 |
| 3 | chr3 | cg11118235 | GNAI2 | 0.05 (0.03, 0.08) | 0.63 | chr8 | cg21847720 | MYOM2 | -0.23 (-0.40, -0.06) | >0.99 |
| 4 | chr14 | cg12154261 | TDRD9 | 0.06 (0.03, 0.09) | 0.63 | chr8 | cg10596483 | JRK | -0.22 (-0.38, -0.05) | >0.99 |
| 5 | chr2 | cg06695611 | ZNF385B;MIR1258 | -0.11 (-0.16, -0.06) | 0.63 | chr13 | cg20293942 | | 0.21 (0.03, 0.38) | >0.99 |
| 6 | chr9 | cg15850063 | | 0.04 (0.02, 0.06) | 0.63 | chr1 | cg08477332 | S100A14 | -0.19 (-0.33, -0.04) | >0.99 |
| 7 | chr3 | cg17241937 | C3orf26;FILIP1L | 0.05 (0.03, 0.08) | 0.63 | chr8 | cg24634471 | JRK | -0.18 (-0.35, -0.02) | >0.99 |
| 8 | chr1 | cg08867825 | OLFM3 | 0.08 (0.05, 0.12) | 0.63 | chr21 | cg00159953 | COL6A2 | 0.18 (0.03., 0.34) | >0.99 |
| 9 | chr8 | cg16903025 | FBXO32 | -0.07 (-0.11, -0.04) | 0.63 | chr6 | cg07185983 | | 0.18 (0.07, 0.29) | >0.99 |
| 10 | chr1 | cg16274353 | TROVE2 | 0.06 (0.03, 0.09) | 0.63 | chr6 | cg25399239 | | 0.18 (0.06, 0.29) | >0.99 |

* Adjusted for multiple comparisons using Benjamini-Hochberg method

^ Model included as covariates maternal BMI (continuous), parity (0 vs 1+), age (continuous), smoking status, quintile of socioeconomic disadvantage, study centre, infant sex, and array batch.

## Discussion

In our investigation of DNA methylation related to an antenatal diet and lifestyle intervention, and overweight and obesity in pregnancy, we have found no evidence of any effect of these factors on DNA methylation in cord blood. In both the main analysis model and a range of sensitivity analyses, we consistently found *no* differentially methylated probes even with a less strict method of Type I error control. Moreover, observed effects were small in magnitude and not consistent in direction.

While a few statistically significant differentially methylated probes were found with data processed using ComBat, and in data normalised using BMIQ, there are reasons to doubt these results. Firstly, the logFC estimates for these probes were extremely small, and (as noted) the significant probes in the BMIQ-normalised data did not appear in 'top 10' probe lists in data normalised using SQN or SWAN. Secondly, implementation of the ComBat procedure allows the user to specify the factors of interest (which in this case were given as intervention group, BMI category and their interaction). Nygaard et al. [29] caution that this may produce spurious effects in situations where the groups are not evenly spread across batches, as was the case in these data. Nevertheless, the discrepancies resulting from different data-processing choices are concerning, and are discussed further in a companion paper (submitted for publication) in which they are investigated more systematically.

### Strengths and limitations

This study has a number of strengths, including its moderately large sample size (645 samples) giving substantial statistical power to detect meaningful differences. Further, these data are from a randomised study with BMI category (25.0–29.9 vs $\geq$30.0 kg/m$^2$) as a stratification variable which was reliably measured in early pregnancy by research staff (rather than self-reported), and are therefore less subject to measurement error or reporting bias. Additionally, participants all had early pregnancy BMI $\geq$25.0 kg/m$^2$, providing greater power to investigate effects of higher BMI, which is often underrepresented in random samples of the population.

The limitations of the study include the study population, the use of cord blood to assess DNA methylation, and the limited coverage of the Illumina 450K array. As the LIMIT study recruited only women with early pregnancy BMI $\geq$25.0 kg/m$^2$, we did not capture the entire BMI range and in particular do not have DNA methylation levels for women of 'normal' BMI. It is possible that there is a nonlinear effect of BMI on DNA methylation, such that the main differences are between women with 'normal' BMI and those with higher-than-'normal' BMI. However, it seems unlikely that there would be substantial differences between women with BMI <25.0 and women with BMI $\geq$25.0, but none between women with BMI 25.0–29.9 and women with BMI $\geq$30.0. We are currently investigating DNA methylation in infants born to participants in the OPTIMISE study (a randomised controlled trial of an antenatal diet and lifestyle intervention for women with early pregnancy BMI 18.5–24.9 kg/m$^2$) to further evaluate the effect of maternal BMI.

Secondly, DNA methylation in cord blood may not be a reliable proxy for the DNA methlation status of infant tissues. Cord blood is commonly used for DNA methylation studies, as it can be obtained non-invasively and in larger quantities [13]. Further, DNA methylation in cord blood is considered to be a good indicator of DNA methylation in infant blood and other tissues [2, 6, 16]. Additionally, cord blood contains different cell types, which may be present in differing proportions in different samples, potentially confounding the effects of interest [36]. While all analyses were adjusted for estimated cell type proportions, the true cell type proportions in the samples are unknown.

Thirdly, the Illumina 450k array analyses around 485,577 sites in the human genome, with a focus on areas of epigenetic interest, i.e., genes and CpG islands [37]. However, this array covers only approximately 2% of CpG sites in the human genome [38]. It is therefore possible that diet and lifestyle in pregnancy, or early pregnancy BMI, have effects on DNA methylation in areas of the genome not covered by the 450K array. Additionally, it is possible that other epigenetic effects may exist, and may interact with DNA methylation. For example, it has been noted that histone modifications may play a part in adipogenesis and hence in susceptibility to obesity [14].

Finally, a larger sample size may be required to reliably detect differences in DNAm due to antenatal interventions, or maternal early pregnancy BMI; the lack of statistically significant findings in the present study may reflect insufficient sample size. However, while a larger sample size would allow detection of smaller differences in DNAm, it is not clear that very small differences would be clinically meaningful.

## Consistency with the existing literature

Our findings may seem at odds with the existing literature, in which numerous studies have found associations between DNA methylation in cord blood, and maternal early pregnancy BMI / obesity. A range of genes and/or loci found to be differentially methylated in relation to maternal obesity and/or BMI have been summarised in S1 Table. These loci include the promoter region of *PPARGC1A* [19]; sites on *ESM1* and *MS4A3* [16]; 86 CpG sites found by the PACE consortium [39]; 28 CpG sites found in the ALSPAC cohort [38]; multiple CpGs mapped to *TAPBP* [39]; a single CpG site mapped to *ZCCHC10* [40]; sites mapped to *FLJ41941* and an unnamed gene [41]; DMRs related to imprinted genes *PLAGL1* and *MEG3* [42]; sites on *MEST* [20]; and 2 CpGs mapped to *RXRA* [10]. Related findings include differential methylation in cord blood in genes *ATP5A1, MFAP4, PRKCH, SLC17A4* related to Gestational Diabetes (GDM) [43]; and hypermethylation of the *LEP* gene promoter associated with maternal obesity on the fetal side of the placenta [44].

However, as indicated by the diversity of this list, the findings from different studies are not consistent, with each study discovering a different set of differentially methylated sites. Where studies have found a range of differentially methylated loci, these are often single CpG sites located on diverse regions of the genome with no known connection to adiposity, obesity, or growth [38, 39]. Moreover, explicit attempts to replicate the findings of other studies have not thus far succeeded [39, 42, 45], and where evidence of differential methylation is found, it is often reported that the actual effect sizes are both of small magnitude, and uncertain clinical significance [18, 39, 41, 43]. Differential methylation may also be found for one analysis approach but not another, e.g. significant findings may become non-significant when analysing BMI as a continuous variable rather than as categories [38]; when adjusting for multiple comparisons [18]; or when adjusting for potential confounders [43]. This lack of consistent, robust evidence has already led others to conclude that DNA methylation is likely not a major causal pathway linking maternal and child obesity [39, 41], with which our findings are in agreement.

Even if reliable evidence of differential DNA methylation in neonates related to maternal obesity / BMI were discovered, it would still remain to be shown that cord blood DNA methylation is causally linked to childhood adiposity, obesity or cardiometabolic health. Some evidence exists that cord blood DNA methylation is associated with child or adult BMI [6, 10, 17, 46]. However, others have found at best weak associations and remain skeptical [14, 38, 47].

## Conclusions

Our study found no evidence of any differentially methylated sites associated with an antenatal lifestyle intervention, or maternal early pregnancy BMI, in cord blood. Moreover, we were unable to find evidence of differential methylation associated with the intervention, or with BMI, for selected candidate genes. The lack of association persisted for different analysis approaches (adjusting for confounders vs not adjusting; using categorical vs continuous BMI; including interaction terms) and for data processed using different methods. Together with the lack of consistent findings from other studies, our results suggest that other causal pathways are primarily responsible for the link between maternal and child obesity.

## Supporting information

**S1 Table. Genes/loci reported as differentially methylated in cord blood in previous studies.**
(DOCX)

**S2 Table. Results of sensitivity analyses.**
(DOCX)

**S1 File. Results of candidate gene analyses.**
(DOCX)

## Acknowledgments

We wish to acknowledge the women who participated in the LIMIT Randomised Controlled Trial and their infants and staff at the participating institutions.

We acknowledge Angela Newman for management of biobank and Dr Jimmy Breen, Bioinformatician/Computational Biologist, for support and guidance with data management.

## Author Contributions

**Conceptualization:** Jennie Louise, Berthold Koletzko, Julie Owens, Jodie M. Dodd.

**Data curation:** Andrea R. Deussen, Jodie M. Dodd.

**Formal analysis:** Jennie Louise.

**Funding acquisition:** Jodie M. Dodd.

**Methodology:** Jennie Louise, Berthold Koletzko, Julie Owens, Richard Saffery, Jodie M. Dodd.

**Project administration:** Andrea R. Deussen, Jodie M. Dodd.

**Resources:** Berthold Koletzko, Jodie M. Dodd.

**Writing – original draft:** Jennie Louise, Jodie M. Dodd.

**Writing – review & editing:** Jennie Louise, Andrea R. Deussen, Berthold Koletzko, Julie Owens, Richard Saffery, Jodie M. Dodd.

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
