## [Decision Letter · Decision Letter 0]

14 Feb 2022

PONE-D-21-39849Effect of an antenatal diet and lifestyle intervention and maternal BMI on cord blood DNA methylation in infants of overweight and obese women: the LIMIT Randomised Controlled TrialPLOS ONE

Dear Dr. Louise,

Thank you for submitting your manuscript to PLOS ONE. After careful consideration, we feel that it has merit but does not fully meet PLOS ONE’s publication criteria as it currently stands. Therefore, we invite you to submit a revised version of the manuscript that addresses the points raised during the review process.

We look forward to receiving your revised manuscript.

Kind regards,

Diane Farrar

Academic Editor

PLOS ONE

Journal Requirements:

5. Please upload a copy of Figure 2, to which you refer in your text on page 14 and 16. If the figure is no longer to be included as part of the submission please remove all reference to it within the text.

Reviewers' comments:

Reviewer's Responses to Questions

**Comments to the Author**

1. Is the manuscript technically sound, and do the data support the conclusions?

Reviewer #1: Yes

Reviewer #2: Partly

Reviewer #3: Yes

2. Has the statistical analysis been performed appropriately and rigorously? 

Reviewer #1: I Don't Know

Reviewer #2: No

Reviewer #3: Yes

3. Have the authors made all data underlying the findings in their manuscript fully available?

Reviewer #1: No

Reviewer #2: Yes

Reviewer #3: Yes

4. Is the manuscript presented in an intelligible fashion and written in standard English?

Reviewer #1: Yes

Reviewer #2: Yes

Reviewer #3: Yes

5. Review Comments to the Author

Reviewer #1: Interesting piece of research, deepening our understanding of epigenetic mechanisms.

There is an error on Table 1 (Baseline characteristics of participants): the table reports that 85% of the participants were smokers, but the text states the contrary.

Reviewer #2: My major concern is that the study may not have sufficient statistical power to detect the small effect size. The power and sample size were not calculated for this secondary analysis. This raises the question whether the results are really negative or the sample size is not enough to detect it.

Table 1 did you compare the characteristics between the two arms? What tests were used and what are the results? All non-significant?

Tables 2 and 3 are not clearly presented.

Any significant interaction? There is no mention of interaction results. If there is no significant interaction, you can show main effects rather than simple effects shown in Table 2 and 3.

What does it mean “intervention at Mean BMI”. Mean BMI of overall?

“Intervention at +5 BMI” is also confusing. Mean BMI + 5 units?

Reviewer #3: The paper is sound, and the authors worked with sufficient data to support their conclusion.

The statistical analysis is rigorous. Different analytical models were used to justify their preposition. The availability of data upon request is documented in the paper. It is well written and in sound English. The strengths presented and well stated and strong. The weakness of the study is also well stated. The implementation of similar study among similar cohort in women with normal BMI is worth considering.

6. PLOS authors have the option to publish the peer review history of their article (what does this mean?). If published, this will include your full peer review and any attached files.

Reviewer #1: No

Reviewer #2: No

Reviewer #3: No

---

## [Author Response · Author response to Decision Letter 0]

21 Apr 2022

Response to Reviewers

1. Ensure manuscript meets PLOS ONE’s style requirements.

We have updated the manuscript to conform to PLOS ONE style requirements.

2. Please provide additional details regarding participant consent.

Written informed consent to participate in the LIMIT study was obtained from all participants (who were all 18 years of age). Additional written consent was obtained to collect samples of umbilical cord blood at delivery, and participants were informed that this would be used for gene expression research related to the diet and lifestyle intervention, and to weight.

This information has been added to the ethics statement, which (per (4) below) has been moved to the Methods section, lines 135-145).

3. Information in ‘Funding Information’ and ‘Financial Disclosure’ sections do not match.

We have expanded the ‘Financial Disclosure’ statement to match the information in the original manuscript. To conform with PLOS ONE style requirements, we have removed this information from the manuscript itself.

4. Ethics statement should only appear in Methods section

The ethics statement has been moved to the methods section and deleted from ‘Declarations’ (lines 135-145).

5. Upload a copy of Figure 2 or remove reference to it within the text

Apologies for the error; this should refer to the bottom panels of figure 1. We have amended the manuscript accordingly.

6. Include captions for Supporting Information files at the end of the manuscript and update in-text citations to match accordingly

Supporting Information is now divided into three files (S1 Table, S2 Tables and S3 Figures), described in lines 581-587 after the references, and the references to this information in the main text have been updated throughout.

7. Review reference list to ensure that it is complete and correct and does not include retracted papers

We have checked the list of references to ensure correctness and completeness. No references have been retracted.

Reviewer 1

1. Error on Table 1 (baseline characteristics) reports 85% of participants were smokers.

Thank you for pointing out this error; it has been corrected. (Line 212 Table 1)

Reviewer 2

2. Study may not have sufficient statistical power to detect the small effect size. The power and sample size were not calculated for this secondary analysis. This raises the question whether the results are really negative or the sample size is not enough to detect it.

Insufficient statistical power is always a potential explanation where differences are not statistically significant; since statistical significance is a function of sample size, a large enough sample would allow detection of statistically significant differences between two groups even though these may not be clinically meaningful.

Our sample size is larger than many other studies of cord blood DNAm relating to antenatal interventions or maternal overweight/obesity, including those which have found statistically significant differences. It provides 80% power to detect differences as small as 0.2 standard deviations between groups; while in the context of high-dimensional data (and resulting multiple-comparisons issues) the question of statistical power is more complicated, we believe that we were adequately powered to detect robust and clinicdally meaningful differences in DNAm.

We have added a sentence to the discussion of limitations (lines 334-337) to note that statistically significant differences may have been found with a larger sample size, but that these differences would have been of uncertain clinical significance. 

3. Table 1 did you compare the characteristics between the two arms What tests were used and what are the results? All non-significant?

We did not perform statistical tests to compare baseline characteristics at baseline; as noted in the CONSORT (Consolidated Standards of Reporting Trials) Statement Explanation and Elaboration) standards, such tests are not recommended. As the groups were randomised, it is already known that any differences between the groups are due to chance. A statistical test, which estimates the probability that differences as large as those observed would arise by chance, is therefore inappropriate.

Additionally, even outside of a randomised setting, statistical significance is not a valid indicator of the presence of confounding (nor is non-significance a valid indicator of absence of counfounding). A statistically significant difference between groups is not necessarily a confounder of the effect of interest, and a non-statistically-significant difference can still confound the effect of interest.

4. Tables 2 and 3 are not clearly presented. Any significant interaction? There is no mention of interaction results. If there is no significant interaction, you can show main effects rather than simple effects shown in Table 2 and 3.

What does it mean ‘intervention at Mean BMI’? Mean BMI of overall?

‘Intervention at +5 BMI’ is also confusing. Mean BMI + 5 units?

The interaction between intervention and maternal BMI was not statistically significant – we have clarified this in the manuscript (lines 222-223).

However, we disagree that a non-statistically-significant interaction implies that ‘main effects’ can or should be estimated instead. Firstly, the analysis model including the interaction term was the prespecified analysis for this study, and should therefore be reported rather than the results of an analysis undertaken after viewing the results of the main analysis. Secondly, the sample size required to detect interaction effects is usually many times larger than that required to detect the individual effects, and tests of interactions are therefore usually underpowered. If the interaction term is dropped (due to non-significance in an underpowered test) and ‘main effects’ are estimated instead, there is potential for the estimates of effects of interest to be biased. As can be seen from Tables 2 and 3, the ‘top 10’ probes differ substantially for effect of intervention at different values of maternal BMI (and for effect of maternal BMI in the different intervention groups).

The interaction term was between intervention (2 groups) and maternal BMI (as a continuous variable). It was therefore necessary, when presenting results, to state the value of maternal BMI at which intervention effects were estimated. ‘Intervention at Mean BMI’ is the effect of intervention estimated at the mean BMI of the cohort; ‘Intervention at +5 BMI’ is the effect of intervention estimated at mean + 5 kg/m2 BMI. We have added some explanation to the Table 2 notes to clarify this.

---

## [Editor Report · Decision Letter 1]

27 May 2022

Effect of an antenatal diet and lifestyle intervention and maternal BMI on cord blood DNA methylation in infants of overweight and obese women: the LIMIT Randomised Controlled Trial

PONE-D-21-39849R1

Dear Dr. Louise,

We’re pleased to inform you that your manuscript has been judged scientifically suitable for publication and will be formally accepted for publication once it meets all outstanding technical requirements.

Kind regards,

Diane Farrar

Academic Editor

PLOS ONE

---

## [Editor Report · Acceptance letter]

14 Jun 2022

PONE-D-21-39849R1 

Effect of an antenatal diet and lifestyle intervention and maternal BMI on cord blood DNA methylation in infants of overweight and obese women: the LIMIT Randomised Controlled Trial 

Dear Dr. Louise:

I'm pleased to inform you that your manuscript has been deemed suitable for publication in PLOS ONE. Congratulations! Your manuscript is now with our production department. 

Kind regards, 

on behalf of

Dr. Diane Farrar 

Academic Editor

PLOS ONE